# On-Site Sensor Calibration Procedure for Quality Assurance of Barometric Process Separation (BaPS) Measurements

**DOI:** 10.3390/s23104615

**Published:** 2023-05-10

**Authors:** Hannah Munz, Joachim Ingwersen, Thilo Streck

**Affiliations:** Biogeophysics Section, Institute of Soil Science and Land Evaluation, University of Hohenheim, 70599 Stuttgart, Germany

**Keywords:** sensor calibration, calibration quality, linear regression model, sensor noise, error analysis

## Abstract

Barometric process separation (BaPS) is an automated laboratory system for the simultaneous measurement of microbial respiration and gross nitrification rates in soil samples. To ensure optimal functioning, the sensor system, consisting of a pressure sensor, an O_2_ sensor, a CO_2_ concentration sensor, and two temperature probes, must be accurately calibrated. For the regular on-site quality control of the sensors, we developed easy, inexpensive, and flexible calibration procedures. The pressure sensor was calibrated by means of a differential manometer. The O_2_ and CO_2_ sensors were simultaneously calibrated through their exposure to a sequence of O_2_ and CO_2_ concentrations obtained by sequentially exchanging O_2_/N_2_ and CO_2_/N_2_ calibration gases. Linear regression models were best suited for describing the recorded calibration data. The accuracy of O_2_ and CO_2_ calibration was mainly affected by the accuracy of the utilized gas mixtures. Because the applied measuring method is based on the O_2_ conductivity of ZrO_2_, the O_2_ sensor is particularly susceptible to aging and to consequent signal shifts. Sensor signals were characterized by high temporal stability over the years. Deviations in the calibration parameters affected the measured gross nitrification rate by up to 12.5% and affected the respiration rate by up to 5%. Overall, the proposed calibration procedures are valuable tools for ensuring the quality of BaPS measurements and for promptly identifying sensor malfunctions.

## 1. Introduction

The barometric process separation (BaPS) technique is a laboratory method for measuring gross nitrification and respiration rates in soil samples [1]. The BaPS technique “separates” the two microbial processes on the basis of total pressure and partial pressure changes in a gastight isothermal incubation system (Figure 1). The simultaneous measurement of air pressure, temperature, and O_2_ and CO_2_ concentrations allows for the quantification of changes in the total gas molecule number *n* and single gas molecule numbers of O_2_ and CO_2_ over time (µmol h^–1^) and the solving of the system’s gas balance equation:(1)$$\Delta n=\Delta O_{2}+\Delta CO_{2}+\Delta X$$

In Equation (1), *∆X* stands for the sum of gases other than O_2_ and CO_2_ that are involved in the gas balance of the incubation system (mainly N_2_, but also N_2_O, NH_3_, CH_4_ and other trace gases). Because the calculation of the microbial process rates is based on a balancing approach, the accurate measurement of *∆n*, *∆O_2_*, and *∆CO_2_* of Equation (1) is essential [1,2].

The BaPS measuring system (Meter Group (formerly UMS AG), Munich, Germany) includes a fully automated measuring head containing CO_2_, O_2_ and pressure sensors (Figure 1) allowing for a continuous (online) measurement of the three state variables at resolutions as low as one minute. The intensive monitoring of the barometric state inside the incubation chamber provides information on the dynamics of gas-consuming and gas-producing microbial processes [1].

The reliability of the BaPS method and the quality of the results depend explicitly on the quality of the calibration of the BaPS sensors; this means an accurate translation of the sensor signals into concentrations or pressure. Calibration errors affect the resulting rates directly as well as on a superordinate level, because relevant measures and conditioned constants from the rate calculation routines are derived from a combination of different sensor measurements, e.g., measurements of the headspace volume *V_head_*; molecule numbers *n*, *nO_2_*, and *nCO_2_*; and physicochemical CO_2_ dissolution [1,3]. Therefore, accurate sensor calibration must be ensured for every measurement. Regular maintenance of proper sensor performance is essential to ensuring high measurement quality.

UMS recommended an annual technical check and recalibration of the sensors [4]. In 2016, however, during the merger of Decagon Inc. (Pullman, WA, USA) and UMS AG (Munich, Germany) to become the Meter Group, the BaPS system was not transferred to the joint product portfolio. Consequently, regular recalibration of the sensors by the manufacturer is no longer possible. The present study was performed before this merger, and the original motivation for developing an easy, do-it-yourself procedure for an on-site calibration of the sensors was that it could allow flexible and regular checks of the sensor calibrations in between the annual checkups by UMS. After the discontinuation of the manufacturer’s calibration, on-site self-calibration has become the only way for BaPS users to recalibrate their sensors, which underlines the importance of the proposed procedure. With the help of this procedure, we were able to monitor sensor signal performance and stability over time, and we could analyze the influence of calibration variation and shifts on the resulting gross nitrification and respiration rates. In the present paper, we describe the calibration procedures in detail, assess the involved error sources and estimate their potential effect on BaPS results.

## 2. Materials and Methods

### 2.1. BaPS Sensors

#### 2.1.1. Temperature Sensors

The BaPS system uses two temperature probes. One probe measures the absolute air temperature in the headspace (T_head_). The other temperature probe is introduced into a soil core to measure the temperature of the soil sample (T_soil_). The temperature probes are platinum (Pt 1000) resistance thermometers which record the decrease in the electric potential due to temperature-induced resistance changes [4]. Pt sensors have a high accuracy (1/3 DIN B), have a low drift, and are stable over time. Thus, quality control and recalibration of the temperature sensors were not considered to be necessary; that is, the manufacturer calibration was left unchanged.

#### 2.1.2. Pressure Sensor

The air pressure (*P*) in the BaPS chamber is measured with a piezoresistive pressure transmitter with an integrated amplifier and temperature compensation [4]. The absolute pressure is measured based on resistance changes due to the deformation of a silicon membrane, which changes the electric potential. The sensor provides an output signal between 0.4 and 2 V.

Pressure is the key variable of the BaPS system because all the other variables in the BaPS calculations directly or indirectly depend on it. Furthermore, the calibration procedure for the O_2_ and CO_2_ sensors proposed hereafter also depends explicitly on the accurate measurement of pressure changes.

#### 2.1.3. O_2_ Sensor

O_2_ concentrations in the headspace are measured by means of a zirconium oxide (ZrO_2_) sensor. At high temperatures, ZrO_2_ becomes conductive for O_2_ and its resistance changes as a function of O_2_ concentration. Thus, at a constant voltage, the electric current changes accordingly. The O_2_ sensor is heated up to 500 °C and produces a 4 to 20 mA output signal (transferred to V by a high-precision resistor bridge). According to the measuring technique, the signal response is nonlinear and specific for the individual sensor [4].

#### 2.1.4. CO_2_ Sensor

CO_2_ concentrations are measured with an infrared (IR) gas sensor that detects the attenuation of an infrared light beam that results from the selective absorption by CO_2_. This measuring technique is known to be very stable; however, the measurement accuracy specified by the manufacturer was the lowest for this sensor and regular recalibration is explicitly recommended [4]. The CO_2_ sensor delivers a signal between 0 and 2.5 V for a measuring range between 0 and 3 Vol%.

### 2.2. System Settings and Calibration Procedure

#### 2.2.1. System Settings

The electronic interface of the UMS BaPS system facilitates communication between the sensors and the computer. These sensor signals are then converted to the respective state variables corresponding to either pressure, temperature, or gas concentrations, by polynomial calibration functions of the form:(2)$$y=a+bx+cx^{2}+dx^{3}+ex^{4}$$
where *y* is the measured state variable (=measurand), *x* is the measured sensor signal and *a*, *b*, *c*, *d*, and *e* are the calibration parameters [4].

To calibrate a sensor, the raw sensor signal must be recorded. To achieve this, in the “Sensor specification” menu, the parameter *b* was set to unity and *a*, *c*, *d*, and *e* were set to zero. No other sensor setting was changed.

During calibration, sensor signals were logged at an interval of one reading per minute using the “Logging” function in the “Current readings” menu. To maximize temperature stability, the set-point temperature of the thermostat was adjusted to ambient temperature.

The BaPS system displays CO_2_ concentrations in µmol mL^–1^ which can be transformed to concentrations in Vol%. Depending on the chosen unit, the raw signal of the CO_2_ sensor varies, so it is important to define the unit for which the calibration is performed within the “Current readings” menu. In this study, the CO_2_ sensor was calibrated in the unit µmol mL^–1^.

The calibration procedures described hereafter were performed simultaneously on three independently operating BaPS systems in order to compare sensor-specific signal behaviors. Calibrations were repeated 6–8 times between 2012 and 2015.

#### 2.2.2. Pressure Sensor Calibration Procedure

The pressure difference between the inside and the outside of the BaPS headspace was recorded by means of a differential pressure manometer with a relative accuracy of 0.2% (GDH 13 AN, Greisinger Electronic GmbH, Regenstauf, Germany) inserted through the rubber septum. In the beginning of a recording, the pressure inside the BaPS chamber was equalized to ambient pressure by piercing a needle through the septum. At zero differential pressure, the sensor signal was recorded manually from the “Current readings” window at high resolution (every 15 s) over a period of 1 to 3 min (5–10 readings). The signal mean was set to absolute ambient air pressure measured by a barometer (G. Lufft Mess- und Regeltechnik GmbH, Fellbach, Germany). Next, the pressure inside the chamber was manipulated stepwise by introducing or removing air to or from the headspace using a gastight syringe (SL syringe 10 mL, Hamilton, Bonaduz, Switzerland). After each manipulation, the sensor signal was recorded over 3 min and the signal mean was paired with the prevailing absolute pressure inside the chamber, calculated as ambient pressure plus/minus the measured pressure difference. The expected pressure change induced by adding or removing gas from the chamber was computed with the ideal gas law (see below).

#### 2.2.3. O_2_ and CO_2_ Sensor Calibration Procedure

The O_2_ and CO_2_ sensors were calminibrated simultaneously. Initially, the headspace air of the empty BaPS system was replaced by synthetic air gas (20.5% O_2_, 79.5% N_2,_ relative accuracy ±2%, Westfalen GmbH, Münster, Germany) via intensive flushing (at least 30 min at 0.02 L s^–1^, escape via an introduced needle). After complete gas replacement, the pressure inside the BaPS chamber was equalized with the ambient pressure by releasing overpressure through a needle. Subsequently, the headspace volume *V_head_* was determined in triplicate via volume extension [1,4]. Sensor readings in synthetic air were recorded for 10 to 15 min with the “Logging” function (10 readings). Then, a defined gas volume (usually 10 mL) was removed from the headspace using the lockable syringe. The pressure decrease induced by gas removal was recorded to determine the number of gas molecules removed. Next, the removed volume was replaced by a CO_2_/N_2_ gas mixture (20.5% CO_2_, 79.5% N_2,_ relative accuracy ±2%, Westfalen GmbH, Germany). The number of added gas molecules was deduced from the measured pressure increase. After the gas exchange, the sensor signal was again logged for 10 to 15 min. The signal mean was related to the calculated O_2_ and CO_2_ concentrations (see below). When repeating the procedure several times, the O_2_ concentrations inside the chamber decreased stepwise, while the CO_2_ concentrations increased accordingly. The total pressure and N_2_ concentration remained unchanged during the procedure. The relevant measuring range was covered when the maximum voltage of the CO_2_ sensor (2.5 V) was reached.

#### 2.2.4. Calculation of Headspace Gas Composition

The gas composition in the headspace was calculated on the basis of the molecule numbers for every gas exchange by mole balance calculations. The total amount of gas molecules in the system *n* was computed using the ideal gas law:(3a)$$n=\frac{P\cdot V_{head}}{R\cdot T}$$
(3b)$$n_{out}=\frac{dP_{out}\cdot V_{head}}{R\cdot T}$$
(3c)$$n_{in}=\frac{dP_{in}\cdot V_{head}}{R\cdot T}$$
where *R* is the ideal gas constant (8.314 Pa m^3^ K^–1^ mol^–1^) and *T* (K) is the headspace temperature. The number of molecules removed from the system (*n*_out_)—calculated from the pressure decrease after gas removal (*dP_out_*) (Equation (3b))—was subtracted from *n*. Then, the removed volume was replaced by the CO_2_/N_2_ gas mixture and the corresponding number of molecules (*n*_in_)—calculated from the pressure increase after injection (*dP_in_*) (Equation (3c))—was added:(4)$$n_{i+1}=n_{i}-n_{out,i}+n_{in,i}$$
where the subscript *i* indicates the *i*-th gas exchange performed.

In the following, gas concentrations given in Vol% are indicated by [ ]. Initially (*I* = 0), 79.5% of *n* was N_2_ ([*N_2_*]*_0_*) and 20.5% was O_2_ ([*O_2_*]*_0_*), while CO_2_ ([*CO_2_*]*_0_*) was 0%. Changes in the individual pools were calculated using Equation (4) for each gas component individually. The molar fraction of *nO_2,i+1_*, *nN_2,i+1_* and *nCO_2,i+1_* of total *n_i+1_* gives the concentrations of the three components after exchange ([*O_2_*]*_i+1_*, [*CO_2_*]*_i+1_*, and [*N_2_*]*_i+1_*). As CO_2_ sensor calibration requires CO_2_ concentrations expressed in µmol mL^–1^, *nCO_2,i+1_* was related to *V_head_* instead of *n_i+1_* after each gas exchange.

Under idealized conditions of constant temperature, *P_i_ = P_i+1_ = P* and *dP_ou_*_t_ = *dP_in_ = dP*, the concentrations [*O_2_*]*_i+1_*, [*CO_2_*]*_i+1_*, and [*N_2_*]*_i+1_* can be calculated as follows:(5)$${\left[O_{2}\right]}_{n+1}={\left[O_{2}\right]}_{i}\cdot \left(1-\frac{dP}{P}\right)$$
(6)$${\left[CO_{2}\right]}_{i+1}={\left[CO_{2}\right]}_{i}\cdot \left(1-\frac{dP}{P}\right)+{\left[CO_{2}\right]}_{inj}\cdot \frac{dP}{P}$$
(7)$${\left[N_{2}\right]}_{i+1}={\left[N_{2}\right]}_{i}\cdot \left(1-\frac{dP}{P}\right)+{\left[N_{2}\right]}_{inj}\cdot \frac{dP}{P}$$
where [ ]*_inj_* is the concentration of the injected CO_2_/N_2_ gas mixture. To convert CO_2_ concentrations from Vol% to µmol mL^–1^, Equation (6) must be multiplied by $\left(\frac{P}{R\cdot T}\right)$. Gas removal–injection cycles were repeated until the full measuring range was covered ([O_2_]: 20.5–17.5 Vol%, [CO_2_]: 0–3 Vol%).

### 2.3. Assessment of Measurement Errors in Sensor Signal and Measurand

Data pairs collected for calibration contain measurement errors in both assessed variables, *x* and *y*—that is, an error in the sensor signal and an error in the measurand, respectively [5]. These errors may affect the accuracy of the resulting calibration. We quantified these errors separately according to the *Guide to the Expression of Uncertainty in Measurement* (GUM) [6,7] and evaluated their potential effect on the calibration parameters. For this, we considered data of several calibration runs in order to obtain a more general perspective on sensor properties and response behaviors.

#### 2.3.1. Measurement Error in x

The measurement error in x, *me_x_*, also called sensor noise, is the standard deviation (s.d.) of the sensor signal. It denotes the sensor-specific fluctuation in the sensor signal under constant conditions (Figure 2) and depends on the specific measuring system and the individual sensor. Furthermore, it influences the calibration data points in the x direction and was quantified over the calibration range by recording the raw sensor signal at each calibration step at constant conditions for several (approx. 10) minutes.

#### 2.3.2. Measurement Error in y

The measurement error in *y*, *me_y_*, arises from the measuring procedures applied to derive the respective measurands (Figure 2). It represents the conjunction of the individual errors of the variables that are involved in the determination of the measurand value [7]. It also influences the calibration data points in the y direction and depends on the accuracy levels of the calibration gases and the measuring devices utilized [5].

In the case of the pressure measurements, *me_y_* depends solely on the accuracy levels of the utilized manometer, which was specified as a relative uncertainty of 0.2% of the measured pressure difference. No further measurement error was involved. Because the pressure variation in the calibration procedure is small (920–1030 hPa), we used the averaged *me_y_* as a constant measurement error over the calibration range.

To estimate the *me_y_* of the O_2_ and CO_2_ concentrations that result from the repeated gas removal and injection cycles, we used a Monte Carlo (MC) simulation approach [7,8]. To set up the simulation spreadsheet (MS Excel, 2010), every input variable of the calculation routine for O_2_ and CO_2_ concentrations according to Equations (3a)–(3c) and (4) (i.e., *V_head_*, *P, dP_out_, dP_in_, T, T_out_, T_in_*) was exchanged by a random number taken from a normal distribution using the built-in Excel function NORMINV, specified by the mean and standard deviation. In the case of the parameters *dP_out,i_* and *dP_in,i_*, *se_b_* of the P calibration was additionally considered. One thousand calibration runs were simulated using a macro sequence. The standard deviations of the resulting gas concentrations were considered as the *me_y_* for O_2_ and CO_2_.

Here, we assumed that the two calibration gas mixtures (O_2_/N_2_ and CO_2_/N_2_) exactly exhibited the concentrations given by the filler in the calibration certificate, i.e., to this point systematic errors were ignored, and only random errors for the procedure-related input variables were considered.

## 3. Calibration Parameterization

According to Equation (2), the BaPS sensors require calibration functions in the form of a linear or polynomial regression, where the sensor signal corresponds to the *x* variable and the measurand (i.e., gas concentration or air pressure) corresponds to the *y* variable.

We used the ordinary least squares method in SigmaPlot version 11.0 (Systat Software, San Jose, CA, USA) to obtain the linear regression parameters *a* and *b*, i.e., the intercept and slope of the calibration line, and the corresponding standard errors of the parameters *se_a_* and *se_b_*. The 95% confidence intervals (*CIs*) of the regression parameters were calculated as
(8a)$$CI_{a}=a\pm t_{crit}\cdot se_{a}$$
(8b)$$CI_{b}=b\pm t_{crit}\cdot se_{b}$$
where *t_crit_* is the critical value of the *t*-distribution at a 95% confidence level (two-tailed value) for n*−2* degrees of freedom (*df*) [9]; i.e., *t_crit_* = 2.37 at *df* = 7.

Applying standard linear regression presumes that the error in *x* is negligibly small. The effect of a failure of this assumption was tested by applying the normal functional model to the data [10]. This model expects a law-like relationship underlying the observed data of the sensor signal and measurand [11,12]. The basic equations are given by
(9a)$$y_{i}=a+bx_{i}+e_{i}\;\mathrm{with}\;i=1,\;\ldots \;k\;\mathrm{and}\;e\sim N\left(0,\sigma^{2}\right)$$
where x*_i_* denotes a fixed but unknown variate and *e_i_* is the additive random and independent variation of *y_i_* (=residuals of *y_i_*) deriving from a normal distribution with a common variance *σ^2^*.

*x_i_* is related to the observed variate *z_ij_* by
(9b)$$z_{ij}=x_{i}+g_{ij}\;\mathrm{with}\;i\;=\;1,\;\ldots \;,\;k\;\mathrm{and}\;j=1,\;\ldots ,\;l\;\mathrm{and}\;g_{ij}\sim N\left(0,{\sigma_{z}}^{2}\right)$$
where *g_ij_* denotes the random variation of *x_i_*. Here, *i* denotes the number of *x_i_–y_i_* pairs in the regression (here, nine), whereas *j* stands for the number of replicated sensor measurements (here, five or ten). The statistical parameters estimated are *a*, *b*, *σ^2^*, *σ_z_^2^*, and the nine *x_i_*. The estimation of the latter may seem unusual but is related to the assumption basic to the functional setting that *x* is a nonrandom variate. The parameters were estimated by using the *nlmixed* procedure of the SAS 9.2 software (SAS Institute Inc., Cary, NC, USA), which employs a maximum likelihood approach.

For each of the three sensors, the resulting calibration parameters *a* and *b* were identical to the parameters resulting from standard linear regression analysis. We concluded that the *me_x_* was in all cases negligible, justifying the use of standard linear regression.

To assess the appropriateness of the linear model, the residuals of the linear model (*e_i_*) were checked for trends [9]. The residual analysis was performed on all available calibration runs as a general test of the linearity of sensor signal response. Additionally, Durbin–Watson coefficients were determined for each sensor on all available *e_i_* to detect autocorrelation.

The residuals *e_i_* are a measure of divergence of the measured data from the functional relationship in the *y* direction [11]. They combine the effects of known (*me_y_*) and unknown (*t*) measurement errors in *y* as well as of a possible error in Equation [13]. The variance of *e_i_*, i.e., σ^2^, describes the variation of the measured data points around the calibration line. The unknown error component *t* can be obtained from the difference in variance:(10)$$t=\sqrt{\sigma^{2}-m{e_{y}}^{2}}$$
which clearly cannot be separated further. *t* is inherent to the measurement and is also called the “individual part” of an observed quantity [14]. (Note that errors in *x* would also increase *σ^2^* [13]. However, the concordances of the functional and linear relationships show that the contribution of *me_x_* to *σ^2^* was negligible.)

### 3.1. Systematic Calibration Errors

Until now, we have only considered the influence of random errors of the variables on the calibration line. The composition of the calibration gases, however, may cause a systematic error for the calibration parameters [8]. We estimated this systematic effect considering the 2% relative accuracy error indicated by the filler (*gasE*). By that, we obtained the maximal range of O_2_ and CO_2_ concentration errors induced by calibration gas inaccuracies, which allowed the estimation of a maximal effect on calibration parameters.

### 3.2. Validation of O_2_ and CO_2_ Calibration

The obtained calibration functions for the O_2_ and CO_2_ sensors were cross-checked against two reference gas mixtures of known gas concentrations. We used a reference gas A with a composition of 17.6% O_2_ and 1.49% CO_2_ and a reference gas B with a composition of 19.7% O_2_ and 0.50% CO_2_, with the remaining percentages for both compositions made up of N_2_ (Westfalen GmbH, Germany). Both gases had a specified relative accuracy of 2% for each component.

The empty BaPS chamber was flushed consecutively with the two reference gas mixtures. The raw sensor signals were recorded for 15 to 20 min as well as the measured O_2_ and CO_2_ concentrations using the latest calibration parameters. The signal means were related to the known gas concentrations of the respective reference gas, and the resulting slopes were compared with the calibration slopes using a paired *t*-test at a significance level of α = 0.05.

### 3.3. Signal Stability

We analyzed several calibration functions determined between 2012 and 2015 in order to recognize general parameter variability and to identify signal drifts or sensor malfunctions. Two indicators of signal stability were adduced: signal strength stability and sensor response stability.

### 3.4. Effect of Calibration Variability on BaPS-Derived Turnover Rates

Finally, the effects of the different calibration errors and calibration functions on the BaPS results (respiration and gross nitrification rates) were analyzed by applying them on the raw sensor readings of an exemplary BaPS incubation. The exemplary BaPS incubation was performed with a silty loess soil from an arable field on the Filder plateau in southwest Germany (Steckfeld, 48°3.1′ N, 9°11.5′ E). The soil pH (water) was 7.76, and the organic carbon content was 0.8% (by weight). The carbonate content was 0.8% (by weight). BaPS calculations on this soil were performed using an adopted respiratory quotient of 0.84 and an experimentally determined CO_2_ dissolution capacity of 0.33 mmol L^–1^ soil solution [15].

### 3.5. Data Availability

All the data analyzed or generated during this study are included in the text, figures, and tables of this article. Complete calibration datasets or further details on the MC simulations and turnover rate calculations are available from the corresponding author on request.

## 4. Results and Discussion

### 4.1. Measurement Errors

#### 4.1.1. Measurement Errors in x

Analysis of several calibration runs revealed that all three sensors showed slight variations in the sensor signal, i.e., *me_x_*, along the calibration ranges. The pressure sensor showed a mean *me_x_* level of 0.2 mV, which corresponded to a pressure error of 0.05 hPa. The O_2_ sensor had a mean *me_x_* level of 0.15 mV, which was equivalent to a concentration error of only 0.002 Vol% O_2_. The mean *me_x_* level of the CO_2_ sensor was 7 mV and was thus higher than the noise levels of the other two sensors. This accounted for a concentration error of 0.005 µmol mL^–1^ CO_2_.

The *me_x_* level of a sensor defines the maximal attainable precision of a sensor reading and, therefore, conditions the limit of detection for changes in the respective measurand [9]. In order to reliably distinguish a measured change from *me_x_*, e.g., during a BaPS measurement, the measured change should exceed three times the noise level [9]. This should be considered in the case of CO_2_ measurements, as *me_x_* is relatively high and concentrations are generally rather low.

In conjunction, the sensor noise levels of the five BaPS sensors (P, O_2_, CO_2_, T_head_ and T_soil_) define a general detection limit of the BaPS method for microbial turnover rates.

In order to use the standard linear regression model with the calibration data, *me_x_* needs to be sufficiently small. As the analysis of the calibration data sets with the mixed-model procedure confirmed that the measurement errors in *x* did not affect the resulting calibration coefficients, this requirement was considered to be fulfilled.

#### 4.1.2. Measurement Errors in y

The *me_y_* of *P* originating from the calibration procedure is equivalent to the relative error of the manometer measurement of 0.2%. This led to the linear increase in *me_y_* with increasing pressure difference (*dP*). The mean *me_y_* of the considered calibration range was 0.052 hPa. The relative uncertainty of a measured *dP* was very low and dropped below 1% at a measured pressure difference of 6 hPa. Thus, the measured values used for calibration can be considered as highly accurate.

The *me_y_* of O_2_ and CO_2_ concentrations were estimated using MC simulations that consider the influence of the stochastic measurement errors of the input variables of the gas balance for each gas exchange (i.e., calibration step). For the entire calibration, we obtained a mean *me_y_* value of 0.0033 Vol% for the derived O_2_ concentrations and 0.0013 µmol mL^–1^ for the derived CO_2_ concentrations. These values can be considered very low as they accounted, on average, for only 0.02% of the measured O_2_ concentrations and for 0.3% of the measured CO_2_ concentrations. These results indicate that the proposed calibration procedure based on repeated gas exchanges allows a precise setting of O_2_ and CO_2_ concentrations within the BaPS chamber (neglecting potential *gasE*, contaminations or other artifacts during the procedure). The high precision of the calculated concentrations can be attributed to the stable incubation conditions (concerning P and T) and the high accuracy of the *dP* measurements.

#### 4.1.3. V_head_

*V_head_* is an input variable of the O_2_ and CO_2_ calibration procedure as well as in BaPS calculations in general. It is derived from the measurement of pressure change due to volume extension, so its accuracy depends directly on the accuracy of pressure calibration. According to the BaPS manual, the achievable relative accuracy for a pressure change measurement of the implemented pressure sensor is of 0.3–0.5% [4]. Including a syringe error of 1%, UMS specifies the relative uncertainty of *V_head_* as 2%. Our estimations for *V_head_* uncertainty via Gaussian error propagation [5,6] of the syringe error and *se_b_* of the pressure calibration resulted in a standard deviation of *V_head_* of approx. 10 mL, which corresponded, at a total chamber volume of 1000 mL, to about 1% relative uncertainty and was mainly conditioned by the syringe error.

### 4.2. Sensor Calibration

In the following section, data and results of the linear regressions of one exemplary calibration per sensor (July 2015) are presented. The residual plots, however, show *y* deviations for all available calibration data sets collected between 2012 and 2015, in order to obtain a general perspective on the linearity of sensor signal response.

Plotting the three different measurands—pressure, O_2_ concentration, and CO_2_ concentration—against the recorded sensor signals reveals the strong linear relationship between the variables for each of the three data sets with coefficients of determination very close to one. Additionally, the standard errors of the calibration coefficients, especially the slope errors *se_b_*, were very low (Figure 3a,c,e), accounting for only 0.42%, 0.36%, and 0.39% of *b* for P, O_2_, and CO_2_ calibrations, respectively. Because BaPS calculations are based on measurand change rates, the standard error of the calibration slope *b* is the most important indicator for calibration quality. The standard error of the calibration coefficient *a* is, in this case, of minor importance.

MC simulations additionally allowed for the estimation of the impact of the *me_y_* errors on the calibration coefficients of the O_2_ and CO_2_ sensors. Those were even lower than the standard errors of the linear regression parameters (Table 1a), indicating that the effectiveness of calibration was not limited by the proposed procedure [12].

Further proof for the calibration effectiveness was obtained by comparing calibration coefficients with company provided sensor calibrations. The pressure and CO_2_ calibration coefficients (Figure 3, Table 1a) coincided almost perfectly with the UMS calibration coefficients (provided by UMS AG in May 2014): P: *a* = 699.9, *b* = 250.08; CO_2_: *a* = −0.01882, *b* = 0.51523. The UMS parameters for the O_2_ sensor (*a* = −4.70367, *b* = 15.58962), however, deviated significantly from the presented calibration coefficients. Therefore, additional calibration validation measurements with two reference gases were performed. These confirmed the obtained calibration parameters of the O_2_ and CO_2_ calibration. The calibration slope of the CO_2_ calibration showed a very good agreement with the reference slope (Figure 3e). The slope difference of 0.006 Vol% V^−1^ was not significant at a significance level of α = 0.05.

In the case of O_2_, the difference between the calibration and the reference slope was 0.13 Vol% V^−1^, which was rather high and at the edge of significance at a significance level of α = 0.05 (t = 2.388, t_crit_ = 2.365). However, we consider the concordance as rather good since the difference between calibration and the reference slope was considerably smaller than the difference between reference slope and UMS slope coefficient (0.65 Vol% V^−1^).

For both sensors, the differences between the calibration and reference slopes fell within the confidence limits of *b* (cf. Figure 3c,d).

Despite the differences in the coefficients, postulated concentration differences between the two reference gas concentrations, *d*O_2_ of 2.1 Vol% and *d*CO_2_ of 1 Vol%, were well reproduced by the calibrated sensors (<0.01 Vol%). This allows for the conclusion that the proposed calibration correctly reproduces concentration changes and that it is thus suited for BaPS measurement applications.

The calibration procedure operates in a dry atmosphere. During a typical BaPS measurement, however, the headspace atmosphere is humid due to the presence of the soil water phase. We tested the effect of humidity on the calibration by adding a water phase to the incubation chamber and the calibration gases. We did not find any systematic differences between wet and dry calibrations. The disadvantage of a wet calibration is that equilibration between the water phase and headspace atmosphere needs several hours. Therefore, and for convenience, we recommend performing the calibration under dry conditions.

### 4.3. Residual Analysis

Although calibration data showed clear linear trends, we analyzed the residuals of several calibration runs to confirm the linear relationship as the general sensor response. If the linear model correctly reflects the sensor response, residuals are expected to scatter randomly along the *x* axis. We found this expectation to be fulfilled for all three sensors. The Durbin–Watson coefficients indicated that the *e_i_* of the calibration data of all three variables were not auto-correlated. As no systematic trends were identified in the residuals (Figure 3b,d,e), we considered the linear regression to be appropriate and sufficient for sensor calibration. This is especially interesting in the case of the O_2_ sensor, where a nonlinear sensor response is stated by the manufacturer [4,16]. Calibration with a linear function has the advantage that a simple a posteriori adjustment of measurand data is possible when calibration deviations are detected.

### 4.4. Variance of the Residuals

Considering the residuals *e_i_* as a measure of the y divergences, these can be used to calculate *σ^2^*, i.e., the variation in the data points around the calibration line. The error *me_y_* explained only 2%, 11% and 20% of *σ^2^* of the P, O_2_ and CO_2_ calibration data, respectively (Table 2), which means that the random variability *t* of *y* (compare Equation (10)) accounts for a major part of the residual scattering. One should recall that *t* represents genuine variability of the experimental material and measurement individuality [11]. Thus, a reduction in the variance and improvement of calibration accuracy is not possible by way of calibration procedure adjustments, but instead depends on the measuring system and inherent variability.

However, the residuals scatter within the calculated 95% confidence limits (Table 2). This, together with the high coefficient of determination of the regression, leads us to the conclusion that, overall, residual variances are very low and that the calibration lines reliably predict the respective measurands.

### 4.5. Potential Systematic Error Due to Calibration Gas Inaccuracies

The estimation of the *me_y_* of headspace O_2_ and CO_2_ concentrations proceeded on the assumption that the measured input variables were not affected by systematic errors. Considering potential errors in the composition of the calibration gases of 2% induced deviations in the observed headspace concentrations of 2%. In the case of O_2_, the absolute standard deviation of the headspace concentrations decreased when the number of gas exchange was increased from 0.41 to 0.36 Vol% and would be two orders of magnitude higher than the conservative estimate of *me_y_*. In the case of CO_2_, the standard deviation of the headspace concentrations increased from zero before gas exchange to 0.025 µmol mL^−1^ (0.059 Vol%) after the last gas exchange and would be one order of magnitude higher than *me_y_*. Additionally, the calibration coefficients reflected the 2% relative *gasE* (Table 1a). The induced deviations of the coefficient *b* highly exceeded its confidence limits (Figure 3c,e).

It is clear that the calibration quality depends directly on the accuracy of the utilized calibration gases. As they exert systematic effects on the calibrations of O_2_ and CO_2_, they can only be corrected when exact quantification is possible. Therefore, a possibility for further improving the accuracy of the calibration procedure would be to cross-check the concentration of the applied calibration gases with an independent analytical method such as mass spectrometry.

As validation measurements confirmed the obtained calibration parameterization, we can assume high concentration accuracy for the utilized gases.

### 4.6. Sensor Signal Stability

Sensor signal stability is an important quality indicator for sensor performance and data reliability and, therefore, contributes to quality assurance for BaPS measurements. The two indicators, signal strength and response stability, are equally important in this context. The sensor signal of the piezoresistive pressure sensor showed very high stability in terms of the signal strength as well as signal response (Figure 4a). The sensor showed a low intercept variability of 2.5 hPa, which may be attributable to the measurement errors of the barometer that were used to determine the reference pressure. This variability, however, can be excluded from further discussion, as it has no influence on BaPS rate calculations. Between the years 2012 and 2015, the calibration slopes varied only 1.1% around the average *b* (Table 1b), without a trend that would indicate sensor aging (Figure 4a). Variations were probably induced by differences in the barometric and thermal background conditions or may be inherent to the measuring system. Despite smaller deviations, the operation of the pressure sensor can be considered stable and reliable.

Comparably, the CO_2_ sensor showed rather stable signal strength and responses over time (Figure 4b). The slope parameter varied 6.1% around the average *b* (Table 1b) but did not show indications of signal drift or sensor aging. The rather high variation may be attributed to the various error components but also to inherent fluctuations in the signal response of the sensor. Since sensor calibration is performed in the low concentration range, the resulting coefficients are more susceptible to absolute errors, e.g., due to small contaminations during the procedure. Though recalibration is explicitly recommended for the CO_2_ sensor [4], we found the sensor signal to be rather constant despite the variability of the calibration coefficients.

Repeated P and CO_2_ sensor calibrations with three independent BaPS systems showed similar stability characteristics. The stability analyses confirmed that the pressure and the CO_2_ sensor provide reliable, high-quality data.

Figure 5 shows temporal recalibrations of the O_2_ sensor of three independent BaPS systems and visualizes fundamental differences in sensor signal stability. In all three BaPS systems, the O_2_ sensor showed high variability in terms of signal strength. In BaPS system A (Figure 5a), the signal response to O_2_ changes was rather constant over time, whereas the sensor signal showed high variability in its absolute strength. Fluctuations in the intercept coefficient *a* accounted for 20% relative variability; the slopes, however, varied only 2.6% over three years around the average *b* (Table 1b). Here, variability did not display a temporal trend. The undirected signal shifts may be caused by small variations of ZrO_2_ properties induced by temperature or moisture conditions [4,16]. Thus, general stability can be assigned to the O_2_ sensor of BaPS system A. By contrast, the O_2_ sensor of BaPS system B (Figure 5b) showed a continuous drift in signal strength over time. This effect can be interpreted as an initial phase of sensor aging, and frequent recalibration is required to ensure BaPS data reliability. In BaPS system C (Figure 5c), massive changes in signal strength and response were observed over time. The observed drift clearly indicated severe malfunction of the sensor and the need for sensor replacement. Reliable BaPS measurements were not possible after 2013. The different behavior of the O_2_ sensor signals over time underlines the necessity of a regular check of the proper functioning of the O_2_ sensor. Due to the applied measuring method based on the O_2_ conductivity of ZrO_2_, the O_2_ sensor is especially susceptible to aging, entailing signal shifts.

The issue of aging for ZrO_2_ sensors is well known, and its development over time depends on the chemistry of the cell, operation conditions and environment [17].

### 4.7. Influence of Calibration Errors on Measured Turnover Rates

We evaluated the effects of effective and potential calibration errors on the results of a BaPS measurement, i.e., the effects on respiration and gross nitrification rates. Table 3 shows the absolute and relative effects of different slope and intercept errors inherent to the present calibration (Figure 3 and Table 1).

Calibration errors ascribable to the proposed procedure, i.e., resulting from the linear regression model (*se_a_* and *se_b_*) and from measurement errors of the measurands (*me_y_*), were very low (Table 1) and, thus, exerted small effects on the measured turnover rates (Table 3). The parameter errors induced by *me_y_* were lower than the standard errors of the calibration coefficients; correspondingly, the rates showed likewise smaller deviations. Overall, the calibration errors were very small in comparison with other error sources occurring during BaPS measurements and rate calculations, such as stoichiometric ratios or considered processes [1,3], as well as natural rate variability.

The most important error source of the calibration was related to the accuracy of the calibration gas concentrations. As the *gasE* affected the important *b* parameters by 2%, effects on the turnover rates were clear, especially on nitrification rates (Table 3). The *GasE* calibration errors entailed maximal rate uncertainties for respiration of approx. 2%, and for gross nitrification of approx. 13%. This potential error can be reduced by the use of high-quality calibration gases. These findings point out that ultimate conclusions on the uncertainty of turnover rates depend primarily on the quantification of systematic error sources such as *gasE*.

Overall, the most important uncertainty source is a result of a lack of signal stability and temporal calibration variability. As we concluded that the observed signal shifts were random, system inherent fluctuations and calibration variabilities were responsible for a major part of the uncertainty of the derived turnover rates. Applying six calibrations on the raw sensor signals of an exemplary BaPS incubation resulted in deviations of 5% and 12.6% in the mean respiration and gross nitrification rates, respectively (Table 4).

Based on the influence of a single sensor’s variability, it can be stated that pressure calibration variations hardly affected the calculated turnover rates. The respiration rates were apparently most sensitive to variations in the CO_2_ calibration. The nitrification rates were likewise affected by O_2_ and CO_2_ calibration.

It is worth discussing that the effect of CO_2_ calibration variability on nitrification rates should be a function of the soil pH, since the calculation of nitrification rates depends highly on the critical gas balance term ∆CO_2,aq_ [1,15]. CO_2_ calibration affects the absolute measurement of the CO_2_ partial pressure and CO_2_ concentration changes. Therefore, errors in the CO_2_ calibration directly influence the estimation of the CO_2,aq_ of soils with pH above 6.5 and, thus, affect the critical dissolution rate ∆CO_2,aq_.

## 5. Summary and Conclusions

We successfully developed a procedure for cross-checking and adjusting BaPS sensor calibrations on site in order to guarantee the optimal performance of the BaPS measuring system and assure high measurement quality between company inspections. We showed that, within the considered ranges, calibration data could be evaluated using linear regression models.

The calibration accuracies of O_2_ and CO_2_ sensors were governed by the accuracy of the utilized calibration gases. Intrinsic calibration variability accounted for the greatest source of calibration uncertainty and exerted the strongest effect on BaPS turnover rates.

All three BaPS sensors generally showed a stable sensor response. However, the O_2_ sensor is especially vulnerable to signal drift and sensor aging due to the utilized ZrO_2_ measuring method operating at high temperatures of up to 500 °C [16]. A regular check of proper O_2_ sensor functioning is advisable. The pressure sensor showed the highest signal stability and lowest effect on the measured turnover rates. A major part of the respiration and gross nitrification rate variability was attributable to shifts in O_2_ and CO_2_ calibration parameters; thus, we point out that the assurance of optimal functioning and the calibration of these two sensors are of particular importance for BaPS measurements.

The proposed procedures proved to be effective tools for detecting sensor malfunction and signal drift. Therefore, the procedures contribute to the quality assurance of BaPS measurements.

## Figures and Tables

**Figure 1 sensors-23-04615-f001:**
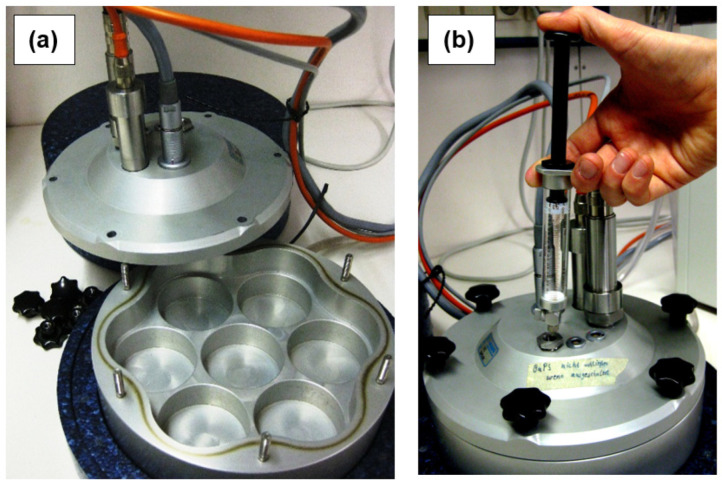
Photographs of the BaPS incubation system. (**a**) Open incubation chamber with cutouts for seven soil cores and measuring head containing the BaPS sensor set consisting of two temperature probes, a piezoresistive pressure sensor, an IR-CO_2_ sensor and a ZrO_2_-O_2_ sensor. (**b**) Closed incubation system with a gastight syringe piercing the rubber septum to introduce or withdraw gas to or from the BaPS headspace.

**Figure 2 sensors-23-04615-f002:**
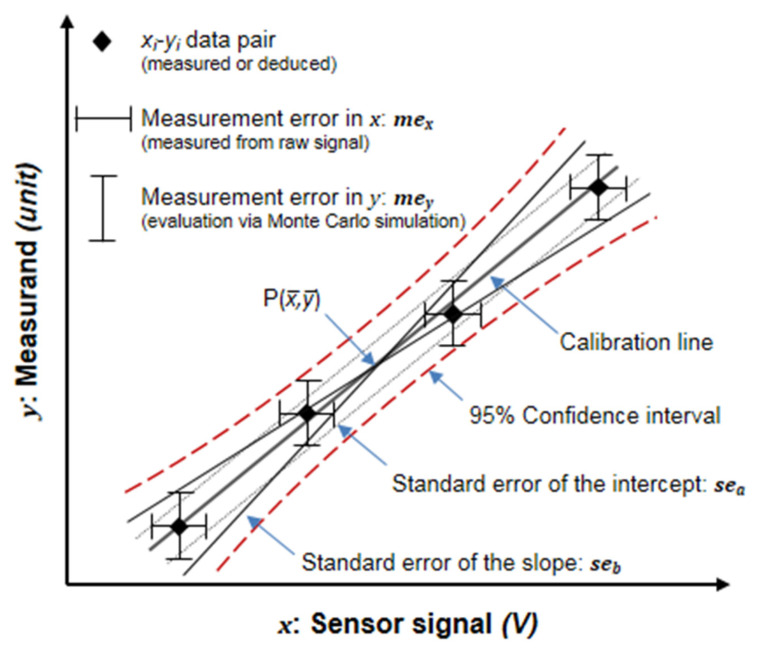
Scheme of a calibration data set (several calibration *x–y* data pairs) and the different error sources involved in the estimation of calibration coefficients: measurement error in *x* (*me_x_*) and measurement error in *y* (*me_y_*). The straight line is the calibration line fitting the recorded *x–y* data pairs according to the ordinary least square method (linear regression), and the dashed lines represent the 95% confidence limits of the entire calibration line. The dotted lines indicate a displacement of the calibration line in the y direction due to the standard error of the intercept (given the slope) (*se_a_*); the thin lines represent the standard error of the slope (given the intercept) (*se_b_*), which consist of rotations of the entire line around $P(\bar{x},\bar{y})$.

**Figure 3 sensors-23-04615-f003:**
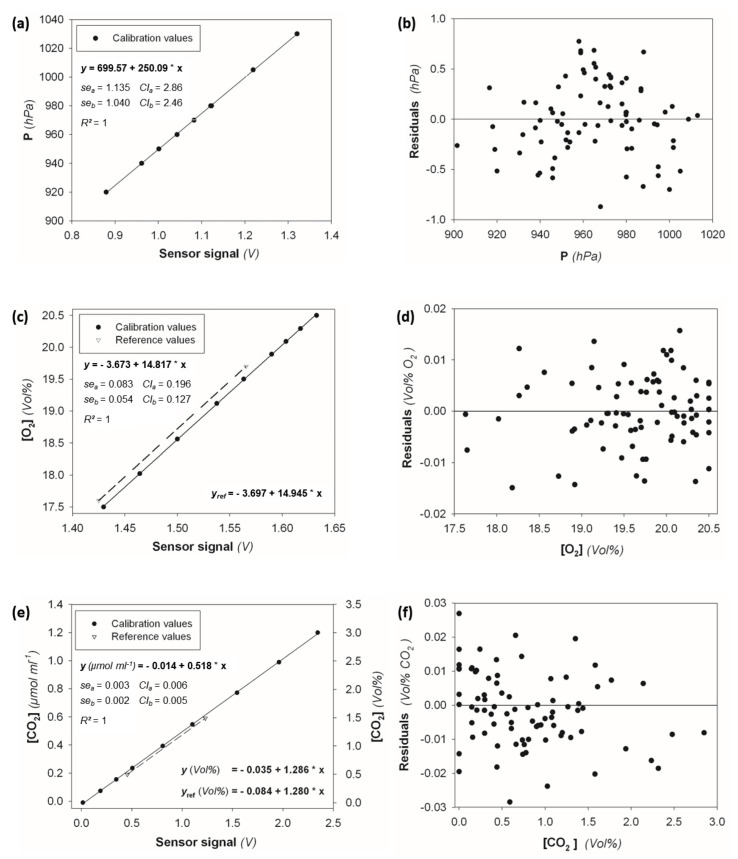
Calibration data, linear regression function, coefficient errors (*se_a_* and *se_b_*), 95% confidence limits (*CI_a_* and *CI_b_*), and coefficient of determination for (**a**) the pressure, (**c**) the O_2_ and (**e**) the CO_2_ sensors, respectively. Triangulars represent the measurement of two reference gases used for calibration validation in (**c**,**e**). Residual plots of six sensor calibrations (linear regression model) performed between 2012 and 2015 are shown in (**b**,**d**,**f**).

**Figure 4 sensors-23-04615-f004:**
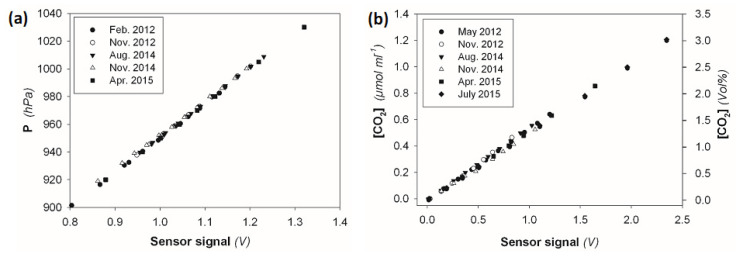
Repeated sensor calibrations between May 2012 and July 2015 for the pressure sensor (**a**) and the CO_2_ sensor (**b**) as well as observed variation of the calibration coefficients *a* and *b*.

**Figure 5 sensors-23-04615-f005:**
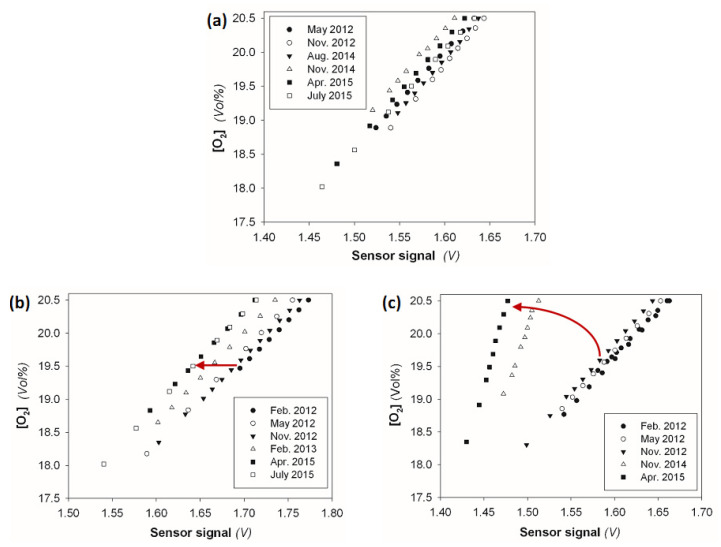
Repeated sensor calibrations between May 2012 and July 2015 for three O_2_ sensors from independent BaPS systems (**a**–**c**). Red arrows indicate the trend of observed drifts in signal strength (**b**) and signal response (**c**).

**Table 1 sensors-23-04615-t001:** (**a**) Calibration coefficient of the July 2015 calibration and a summary of the calibration coefficient errors induced by the different error sources: standard errors of the coefficients (*se*), coefficient deviations induced by measurement errors (*me_y_*), and potential coefficient deviations assuming 2% calibration gas errors (*gasE*). (**b**) Calibration coefficients averaged over the years 2012 to 2015 (n = 6) and their respective variabilities, indicated by the standard deviation (s.d.) of the coefficients.

(a)	P Calibration		O_2_ Calibration		CO_2_ Calibration	
**Calibration July 2015**	***a*** (hPa)	***b*** (hPa V^−1^)	***a*** (Vol%)	***b*** (Vol% V^−1^)	***a*** (µmol mL^−1^)	***b*** (µmol mL^−1^ V^−1^)
coefficient	699.6	250.1	–3.69	14.83	–0.015	0.52
*se*	±1.14	±1.04	±0.08	±0.05	±0.0030	±0.0020
*me_y_*	±0.27	±0.25	±0.03	±0.02	±0.0009	±0.0007
*gasE*			±0.07	±0.30	±0.0003	±0.0100
**(b)**	**P Calibration**		**O_2_ Calibration**		**CO_2_ Calibration**	
**Calibrations** **2012–2015**	***a*** (hPa)	***b*** (hPa V^−1^)	***a*** (Vol%)	***b*** (Vol% V^−1^)	***a*** (µmol mL^−1^)	***b*** (µmol mL^−1^ V^−1^)
mean	699.3	250.6	–4.17	15.13	–0.02	0.54
s.d.	±2.49	±2.64	±0.76	±0.39	±0.01	±0.03

**Table 2 sensors-23-04615-t002:** Mean residual (*e_i_*) and mean 95% confidence interval (*CI*) of the pressure, O_2_ and CO_2_ calibration from July 2015. Decomposition of the residual variance σ^2^ into measurement error component me_y_^2^ and random error component t^2^.

Sensor		Mean *e_i_*	Mean *CI*	*σ^2^*	*me_y_^2^*	*t^2^*
Pressure	(hPa)	0.28	0.37	0.154	0.003	0.151
*O_2_*	(Vol%)	0.008	0.012	116 × 10^−6^	13 × 10^−6^	103 × 10^−6^
*CO_2_*	(µmol mL^−1^)	0.003	0.005	10 × 10^−6^	2 × 10^−6^	8 × 10^−6^

**Table 3 sensors-23-04615-t003:** Effect of the errors *se_a_*, *se_b_*, *me_y_* and *gasE* of the calibration “July 2015” on the derivation of respiration rates and gross nitrification rates. Listed are the deviations from the original rates in absolute and relative terms (the latter are in brackets).

Calibration: July 2015	Respiration Rate (µgc Kg^−1^ Sdw ^†^ H^−1^)	Gross Nitrification Rate (µgn Kg^−1^ Sdw ^†^ H^−1^)
**Original Rates**	**393.25**	**77.59**
Deviation due to *se_a_* and *se_b_*	±1.32 (±0.3%)	±1.51 (±1.9%)
Max. deviation due to *me_y_*	±0.42 (±0.1%)	±0.77 (±1.0%)
Max. deviation due to *gasE*	±7.48 (±1.9%)	±10.20 (±13.1%)

^†^ Soil dry weight.

**Table 4 sensors-23-04615-t004:** Effects of temporal calibration coefficient shifts of the O_2_, CO_2_ and pressure sensors on respiration and gross nitrification rates. Based on the raw sensor data of an exemplary BaPS incubation, the respiration and nitrification rates were computed on the basis of six different calibrations performed between May 2012 and July 2015.

Calibration	Respiration Rates (µgc Kg^−1^ Sdw ^†^ H^−1^)	Gross Nitrification Rates (µgc Kg^−1^ Sdw ^†^ H^−1^)
May 2012	408.3	66.13
Nov 2012	445.7	59.42
Aug 2014	413.7	77.65
Nov 2014	385.9	84.48
Apr 2015	402.4	78.62
July 2015	393.4	77.67
**mean**	**408.2**	**74.0**
** *s.d.* **	**20.9**	**9.3**
***CV* ^‡^ (%)**	**5.1**	**12.6**

^†^ Soil dry weight; ^‡^ coefficient of variation.

## Data Availability

The data presented in this study are available upon request from the corresponding authors.

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
