# Peer review of "On-Site Sensor Calibration Procedure for Quality Assurance of Barometric Process Separation (BaPS) Measurements"

_sensors, 2023, doi:10.3390/s23104615_

Round 1
Reviewer 1 Report
The author theoretically discussed the Barometric Process Separation (BaPS) is an automated laboratory system for simulta- 7 neous measurement of microbial respiration and gross nitrification rates in soil samples. To assure 8 optimal functioning, the sensor system, consisting of a pressure sensor, an O2 and a CO2 concentra- 9 tion sensor as well as two temperature probes, must be accurately calibrated. For regular on-site 10 quality control of the sensors, we developed easy, inexpensive, and flexible calibration procedures. 11 The pressure sensor was calibrated by means of a differential manometer. O2 and CO2 sensor were 12 simultaneously calibrated by exposing them to a sequence of O2 and CO2 concentrations obtained 13 by sequentially exchanging O2/N2 and CO2/N2 calibration gases. Linear regression models were best 14 suited to describe the recorded calibration data. The accuracy of O2 and CO2 calibration was mainly 15 affected by the accuracy of the utilized gas mixtures. Due to the applied measuring method based 16 on O2 conductivity of ZrO2, the O2 sensor is especially susceptible for aging and entailed signal shifts. 17 Sensor signals were characterized by high temporal stability over the years. Deviations of calibrate- 18 tion parameters affected measured gross nitrification rate by up to 12.5% and of the respiration rate 19 up to 5%. Before accepting an author, I have some major concerns that need to be addressed.
1. The authors need to correct several grammatical errors in the present manuscript.
2. The authors discussed more theoretical predictions, it would be good if they could provide some experimental evidence to support their claims.
3. It is necessary for the author to add and explain the purposed setup as well as draw some experimental structures.
4. I suggest the author should calculate his results with both N2 and CO2 combinations as well as N2 alone since the authors discussed more CO2 and O2 combinations. The manuscript should also include those results and discuss them.
5. In each step, the authors must calculate the standard deviation and maximum probable error and include them in their manuscript.
Author Response
Dear Reviewer,
thank you very much for the critical reading of our manuscript “On-site Sensor Calibration Procedure for Quality Assurance of Barometric Process Separation (BaPS) Measurements” and for your suggestions for improvement. We tried to include all your recommendations and suggestions.
- The authors need to correct several grammatical errors in the present manuscript.
We went through the manuscript carefully and corrected grammatical and orthographical errors.
- The authors discussed more theoretical predictions, it would be good if they could provide some experimental evidence to support their claims.
We are not fully clear what you mean with theoretical predictions, but we guess you mean the theoretical calculation of the adjusted gas concentrations in the incubation chamber during calibration. If your suggestion is to check the calculated concentrations against an independent method for means of verification, we fully agree with you. Unfortunately we are not able to cross-check this anymore as the gases we used in the study between 2012 and 2015 are not available anymore. However, due to the accordance between the achieved calibrations and the cross-check with high-quality reference gases, we are confident that concentration calculations are reliable.
- It is necessary for the author to add and explain the purposed setup as well as draw some experimental structures.
We added two photographs of the BaPS measuring system in order to give an impression of the setup and the device (Fig.1).
- I suggest the author should calculate his results with both N2 and CO2 combinations as well as N2 alone since the authors discussed more CO2 and O2 combinations. The manuscript should also include those results and discuss them.
We are very sorry but results of N2-CO2 combinations and N2 alone do not exist. Those combinations were not performed as they were not regarded to be relevant for the calibration. As mentioned above, those measurements cannot be repeated as gases and devices are not available anymore.
- In each step, the authors must calculate the standard deviation and maximum probable error and include them in their manuscript.
Measurement errors in x and y are presented as the standard deviations either of the sensor signal (mex) or of the concentration (mey), obtained by a Monte-Carlo Simulation (p.6, l. 207, p.7, l. 241). Therefore, in our view, we calculated the standard deviations. The maximum (probable) error of the calibration data is, as stated, determined by the systematic error of 2% of the gases used for calibration and validation.
Reviewer 2 Report
The authors presented an article in which they described methods for calibrating pressure sensor, oxygen and carbon dioxide sensors, which are parts of one device. Possible measurement errors and their sources are analyzed in detail. In general, the work is more technical, the scientific novelty is not very clear. In this regard, the following questions and suggestions arise:
1. An oxygen sensor based on ZrO2 and an IR CO2 sensor are very common, long-known sensors. Their parameters and long-term stability have already been analyzed in other works? You should present this information clearly in Introduction, compare then it with your own results.
2. The method of calibrating of O2 and CO2 sensors is trivial. The gas mixtures used for calibration have a composition error of 2%, which, as the authors rightly point out, is a source of systematic error. One could try to analyze the composition of a commercial gas mixture by an independent method (mass spectrometry, for example), which, perhaps, would give a smaller error in the composition and a greater measurement accuracy.
3. Maybe I'm not well versed in the application of the device, but it is possible that in the study of microbial respiration, the chamber is filled with a humid atmosphere while you were calibrating with dry gas mixtures. Does humidity affect the readings of oxygen and carbon dioxide sensors?
4. The quality of Figure 2 is too low, please try to improve it.
5. [9] is a reference to the PhD thesis, is it possible to replace (or add) this with a similar peer-reviewed article?
Author Response
Dear Reviewer,
thank you very much for the critical reading of our manuscript “On-site Sensor Calibration Procedure for Quality Assurance of Barometric Process Separation (BaPS) Measurements” and for your valuable suggestions for improvement. We tried to include all your recommendations and suggestions.
In general, the work is more technical, the scientific novelty is not very clear.
We agree that our study is rather technical than scientific. The relevance and novelty of the study is related to the demand of a on-site method to continue the operation of all BaPS systems around the world. Therefore, we intended to develop a simple and easy but still highly accurate method, so that all BaPS-users can easily use it. We tried to stress this demand and our motivation in the introduction part (p. 2, ll. 60-73).
- An oxygen sensor based on ZrO2 and an IR CO2 sensor are very common, long-known sensors. Their parameters and long-term stability have already been analyzed in other works? You should present this information clearly in Introduction, compare then it with your own results.
Thank you for your suggestion. We tried to find more information on long-term sensor stability of ZrO2-Sensors and the aging process. We did not find precise data but information on general behavior. We included this information in the Discussion part.
-> 15, line 525: This aging issue of ZrO2 sensors is well known, and its development over time depends on the chemistry of the cell, operation conditions and environment (Burkhard et al., 1995).
- The method of calibrating of O2 and CO2 sensors is trivial. The gas mixtures used for calibration have a composition error of 2%, which, as the authors rightly point out, is a source of systematic error. One could try to analyze the composition of a commercial gas mixture by an independent method (mass spectrometry, for example), which, perhaps, would give a smaller error in the composition and a greater measurement accuracy.
Thank you for this very reasonable suggestion. We agree that this kind of analysis would have reduced the systematic error of the calibration. Further, MS measurements of the headspace O2/CO2 concentration after the calibration procedure would have been a good cross-check for the accuracy of the calibration procedure and would have revealed potential contaminations of the system.
Unfortunately, we do not have this kind of data and we are not able to perform these measurements at this point of our work as the devices and gases are not available anymore. We added a sentence in the discussion part promoting this suggestion for a further improvement of the calibration procedure.
-> p. 15 l. 486 ff
- Maybe I'm not well versed in the application of the device, but it is possible that in the study of microbial respiration, the chamber is filled with a humid atmosphere while you were calibrating with dry gas mixtures. Does humidity affect the readings of oxygen and carbon dioxide sensors?
You are very right. The BaPS system operates with wet soil samples and thus, with a humid headspace air. Thus, your question on the influence of water vapor on sensor performance is very valid.
Generally, it is stated by the sensor manufacturer that the ZrO2 sensor is very sensitive to water and humidity and can easily be damaged by condensation water. To avoid condensation, the BaPS system always needs to be ventilated.
We were also aware of this important question, therefore, we performed both, calibrations in dry systems, as presented in the manuscript, as well as calibrations in a wet system where a water phase was included in the closed BaPS system and in the gas storage vessels.
We did not find a systematic difference between wet and dry calibrations neither we observed an influence on the sensor signals, that could be assignable to the influence of humidity on the measurement mechanism. The disadvantages of a wet calibration, however, is that the wet calibration is more time demanding as the equilibration between headspace atmosphere and the water phase requires several hours. Therefore, we decided to work under dry conditions.
We include this information in the manuscript in the discussion part, however, we did not include data.
-> 13, l. 434 – p. 14 l. 441
- The quality of Figure 2 is too low, please try to improve it.
We improved the quality of Figure 2 (now Figure 3, after introduction of a new figure).
- [9] is a reference to the PhD thesis, is it possible to replace (or add) this with a similar peer-reviewed article?
We exchanged the PhD thesis of Dr. Guillard by a peer-reviewed publication of him.
-> see references
Round 2
Reviewer 2 Report
Authors responded exhaustively to the comments, the article can be published in the present form.
Author Response
Thank you very much for your careful revision and for your recommendation.